# Demand Side Management Based Power-to-Heat and Power-to-Gas Optimization Strategies for PV and Wind Self-Consumption in a Residential Building Cluster

**Marcus Brennenstuhl** [1,*] **, Daniel Lust** [1] **, Dirk Pietruschka** [1] **and Dietrich Schneider** [2]

1   Centre for Sustainable Energy Technology (zafh.net), Stuttgart University of Applied Sciences, Schellingstr. 24, 70174 Stuttgart, Germany; daniel.lust@hft-stuttgart.de (D.L.); dirk.pietruschka@hft-stuttgart.de (D.P.)
2   Steinbeis-Innovationszentrum LOCASYS-Innovations, Osterholzallee 140-7, 71636 Ludwigsburg, Germany; dietrich.schneider@hft-stuttgart.de
*   Correspondence: marcus.brennenstuhl@hft-stuttgart.de

**Abstract:** The volatility of renewable energy sources (RES) poses a growing problem for operation of electricity grids. In contrary, the necessary decarbonisation of sectors such as heat supply and transport requires a rapid expansion of RES. Load management in the context of power-to-heat systems can help to simultaneously couple the electricity and heat sectors and stabilise the electricity grid, thus enabling a higher share of RES. In addition power-to-hydrogen offers the possibility of long-term energy storage options. Within this work, we present a novel optimization approach for heat pump operation with the aim to counteract the volatility and enable a higher usage of RES. For this purpose, a detailed simulation model of buildings and their energy supply systems is created, calibrated and validated based on a plus energy settlement. Subsequently, the potential of optimized operation is determined with regard to PV and small wind turbine self-consumption. In addition, the potential of seasonal hydrogen storage is examined. The results show, that on a daily basis a 33% reduction of electricity demand from grid is possible. However, the average optimization potential is reduced significantly by prediction inaccuracy. The addition of a hydrogen system for seasonal energy storage basically eliminates the carbon dioxide emissions of the cluster. However, this comes at high carbon dioxide prevention costs of $1.76 \text{€} \, \text{kg}^{-1}$.

**Keywords:** demand response; demand side management; heat pump optimization; building simulation; HVAC optimization; genetic algorithm; heat pump cluster operation; demand flexibility; small wind turbine

## 1. Introduction

In order to counteract climate change, energy supply systems in all relevant sectors need be decarbonized. The problem thereby is that with regard to the share of RES there is a clear sector imbalance. In Germany, in 2018, on the electricity demand side, RES had a share of approximately 40% [1]. In contrary, the share was only 14% on the heat demand side [2]. To solve this problem sector coupling in the context of power-to-heat can present an important solution. This is underlined by the fact that it is predicted that heat pumps will gain more importance within future heat supply systems [3]. Additionally, in the near future, also hydrogen-based systems (power-to-gas) will play an increasing role for the decarbonization of heat generation in the building sector, especially for existing buildings, where heat pumps are not always suitable.

Within this work, based on a real existing plus energy district as case study, a detailed digital twin of a cluster of 10 residential buildings with decentral heat pumps, thermal buffer storages and large PV systems is created and validated. The buildings are supplied by a cold district heating grid that is fed by an agrothermal collector (a large-scale specialized version of a geothermal collector). In a further step, the system model is supplemented by a

hydrogen system with a PEM-Electrolyzer, local hydrogen storage and a power-driven fuel cell, to increase the autarky of the cluster by storing seasonal power surpluses as hydrogen. Thereby during the heating season, the fuel cell provides additional power for the building heat pumps, as well as waste heat. Based on this digital twin, an optimized single building and cluster operation of the buildings heat pumps and the hydrogen system is realized with the aim to increase PV self-consumption (either through direct usage by heat pumps or by storage as hydrogen and re-electrification), annual heating energy autarky and to provide more flexibility to utilize on site generated electricity from small wind turbines.

In this study, first, the state of the art regarding model-based prediction and optimization algorithms in the context of HVAC systems is presented. Then a detailed description of the model calibration methodology that relies on the application of an optimization algorithm to model parameters in combination with measured data (set point temperatures, air exchange rate etc.) is given. Also prediction inaccuracy due to weather forecast uncertainties are examined. Additionally the optimal parameters for the genetic algorithm to produce 24 h schedules for heat pump operation according to predicted weather data are determined. This ensures a low computational demand and at the same time a high quality of results. Then based on the calibrated cluster model and the optimal algorithm parameters, results for different optimization goals are obtained for an entire year. Thereby a 24 h optimization for each day of the year is carried out based on logged weather prediction data. Those results are then compared to the optimized operation based on the weather situation that has occurred in reality. The potential of the following optimization goals is examined:

- Single building: Optimization of PV self-consumption.
- Cluster: Optimization of PV self-consumption.
- Cluster: Optimization of PV and small wind turbine self-consumption.
- Cluster: Autarky and increased carbon dioxide emission saving with the application of a hydrogen system.

### 1.1. State of the Art

#### 1.1.1. Building Demand Prediction

Advanced prediction, optimization and control algorithms are essential to realize intelligent load management (demand response) in HVAC and especially in heat pump systems as well as to enable an improvement in efficiency. Prediction methods can be divided into three different subcategories: analytical, numerical (e.g., detailed building simulations), and predictive (self-learning algorithms such as neural networks) [4]. Furthermore, the modeling approach can be distinguished between so-called White Box, Grey Box, and Black Box models. White Box and Grey Box models belong to the numerical methods, Black Box models to the predictive methods. White Box building simulation environments such as EnergyPlus, TRNSYS and INSEL tend to provide more accurate results because they combine several domains such as heat demand, internal thermal gains, ventilation [5] and because they represent buildings in a more physical or geometric detail. The disadvantage of these models, especially with respect to applications that should run in near real time, is the higher computational effort, which is usually better with Grey Box and Black Box models. On the other hand, the parameterization of Grey Box models is error-prone and Black Box models require a high data quality over longer periods. An overview over the different methods and their advantages and disadvantages is given in Table 1. Here also a simplified White Box is listed. This refers (i) to the methodology used in this work, that consists of single zone building models with a lower level of geometric detail and (ii) to other more simplified methods.

**Table 1.** Comparison of different building demand prediction methods. Own representation based on [6].

| Method | Required Input Data | Typical Applications | User-Friendliness | Computational Demand | Model Accuracy |
|---|---|---|---|---|---|
| White Box (Detailed) | Detailed physical information | DOE-2, EnergyPlus, TRYSYS, ESP-r INSEL | Low | High | High |
| White Box (Simplified) | Physical information | Heating degree day method, temperature frequency method, RLF (Residential Load Factor) | Average | Average | Relativly high |
| Grey Box | Physical information and historical data | RC Network | Low, average | Low, average | Relatively high |
| Black Box | Historical data | ANN, SVM, Regression-analysis, Clustering | Low | Low, average | High except regression |

### 1.1.2. Metaheuristic Optimization in the Field of HVAC Systems

One significant advantage of metaheuristic algorithms is that in opposite to linear and nonlinear programming the equations on which the mathematical model is based on, don't need to be accessible by the optimization algorithm. This is important when using building simulation tools like EnergyPlus, TRNSYS or INSEL. The drawback is that only a near by global optimum can be achieved. In the case of building demand prediction methods this can be neglected because due to prediction inaccuracies (e.g., ambient temperature and global radiation), the real global optimum will always be slightly different. The field of metaheuristic algorithms can be divided into the subcategories trajectory-based (single-individual algorithms) and population-based [4,7]. Often used examples of population-pased algorithms are particle swarm optimization (PSO), ant colony optimization (ACO) and genetic algorithms (GA). GAs are characterized by the fact that, like various stochastic techniques and population-based algorithms, they are based on nature and, in particular, on the theory of evolution and natural selection. Natural selection, which was largely defined by Charles Robert Darwin, states that the individuals in a population that are more fit or more able to survive tend to reproduce more and thus pass on their genetic material and with that their positive characteristics. Due to their design, GAs are variable and therefor well adaptable to different problems [8].

According to a literature review by Ahmad et al. [4], in the field of optimization of HVAC systems, most applications are based on stochastic algorithms to which the field of heuristic and metaheuristic algorithms also belongs. Most scientific publications aimed at solving HVAC related problems made use of evolutionary algorithms (EA) and GA. Among them, GAs were most frequently used at 37%, followed by error backpropagation in the context of artificial neural networks at 35% and PSO at 19%. In the field of building energy systems, software-based optimization can be useful in different areas, where one can distinguish between the optimization of the system dimensions and the system control. Different population-based algorithms are used for the optimization of HVAC systems. It is shown that EA and GA are especially suitable for the search of a global optimum, but their convergence speed may be lower compared to other algorithms. PSO, on the other hand, showed better performance in searching local optima and the search process is slower compared to GA. Compared to GA, ACO algorithms are also suitable for finding global optima and they show higher convergence speed in searching local optima. GAs are mainly used when no unique solution is sought. They are also usually used for complex optimization tasks because they are better suited for global searches. In some cases, GAs are also combined with artificial neural networks (ANN) as optimization algorithms to model complex nonlinear systems [4,9].

In previous studies, a coupling of demand prediction (based on detailed white box models of the building and energy supply) in combination with optimization algorithms was examined. For example, by Zhou et al. [10] an integrated optimization module for EnergyPlus was developed to use the thermal mass of an office building as energy storage to reduce the peak load of the building air conditioning. For this purpose, schedules of the temperature set points were created for 24 h in advance. Four different optimization algorithms were investigated in this context. The Nelder-Mead-Simplex method (a simpler evolutionary algorithm), the Broyden-Fletcher-Goldfarb-Shanno (BFGS) method, simulated annealing and a population-based approach. Among these, the Nelder-Mead-Simplex method showed the best results. Here, a computation time of 1–1.5 h resulted in 33% savings in power costs. In comparison, the use of GenOpt as an external optimizer led to an increase in computation time to 6 h. The settling time was taken into account using weather data from the previous day. Xu et al. [11] optimized the operation of their university laboratory building using an EnergyPlus model and PSO. The temperature set points were varied to achieve optimal pre-cooling of the building during the following day. This resulted in a 21.3% reduction of energy costs. Kajl et al. [12] used a GA to optimize the set points (air exchange rate, chilled water supply temperature) of a local energy management (LEM) system in a building on an university campus. The set points were optimized for two summer months, resulting in a 16% reduction of energy demand. The computation time for this was 30 min. GA were also used to optimize the thermal building envelope by Sun et al. [13] and to achieve an optimal combination by Seo et al. [14] or sizing by Bee et al. [15] of building energy supply systems. What many of those studies neglect, as in those by Zhou et al. [10] and Xu et al. [11], is to address the effect of weather prediction inaccuracy on the optimization result. This can lead to an overestimation of the optimization potential.

### 1.1.3. Seasonal Hydrogen Storage

The usage of a power-to-hydrogen system as a seasonal energy storage is described by Petkov and Gabrielli [16]. They state that the application of hydrogen systems are beneficial for districts with high thermal-to-electricity demand ratios, where a seasonal hydrogen storage system contributes to shave peak electrical demands in winter. This minimizes the overall carbon footprint and increases the self-consumption of RES of the investigated district. However, according to this study, power-to-hydrogen only contributes to the last 5% to 10% of carbon emission savings at high investment costs.

Wang et al. [17] describe a power control strategy for a wind/PV/fuel cell hybrid energy system for five homes with an electrolyzer as dump-load. The fuel cell is sized to match the peak elecetricity load, whereas the electrolyzer is designed to handle the maximal excess power. This study is limited to the consideration of the electrical load. Heat loads are not regarded, which means that the additional electricity demand caused by heat pumps with their specific load profiles are not part of this study.

The possibilities of optimizing the operation of a hybrid system with a fuel cell and a heat pump for a single residential building is shown by Sun et al. [18]. A day-ahead scheduling is used to minimize the fuel costs, with hydrogen used as a single energy carrier. However, this study focuses on a single building and single day optimization. Electrical power generation from RES is also not considered in this study.

In a study about possible self-sufficiency for single-family houses in central Europe with heat pumps, Knosala et al. [19] showed that self-sufficiency could be achieved with a hydrogen system based on PEM technology and a pressurized hydrogen storage at lower costs than with seasonal storage based on lithium-ion batteries. They outlined that self-sufficiency in building cluster could be more simply achieved compared to their investigated scenario.

## 2. Methodology

### 2.1. Pilot Site

This work focuses on a real existing building cluster situated within a plus energy demonstration site in the German municipality of Wüstenrot. The pilot site consists of 23 recently built residential buildings with high-energy standard (see also Figure 1). All buildings have decentral heat pumps that are supplied by a cold district heating grid that is connected to a large scale low depth geothermal system (agrothermal collector). All buildings are equipped with PV systems between 6 kWp and 28.8 kWp. Within 12 buildings, thermal and electrical energy flows as well as temperatures are monitored in detail. These monitoring data are accessible by a cloud based monitoring and control system. A detailed description of the pilot site and it's novel heat supply system is given by Brennenstuhl et al. [20] and by Pietruschka et al. [21] .

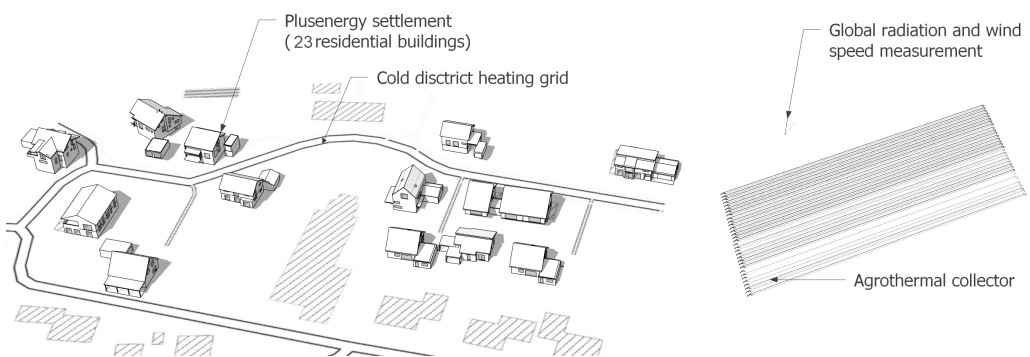

**Figure 1.** Plus energy settlement "Vordere Viehweide".

### 2.2. Modelling Approach

#### 2.2.1. Building Model

Within this work dynamic building models are used that are based on the nodal method. This method is common for building simulations due to its ability to resemble multi zone buildings. Thereby the principal assumptions are, that each zone is a homogeneous volume that is characterised by uniform state variables where a node can e.g., represent a room, a wall or a window. The thermal transfer equations are solved for each node of the system. Each building model is divided into different parts. The connection between them is realized with internal connectors (inputs and outputs for the blocks, see also Figure 2). The model bases on a one dimensional numerical solution of the heat conduction equation for each wall. Additional nodes resemble room air and windows. Long wave radiation exchange between the room surfaces is also taken into account. Inputs for the model are external boundary conditions (temperature and irradiance). To speed up the model creation process and reduce the computational effort, all buildings are resembled as single zone models with a lower level of geometric detail. To counteract a possible reduction of prediction accuracy by the reduced model detail, a calibration method is applied that is described in Section 2.5.

#### 2.2.2. Decentral Energy Supply System

Regarding the heat pumps, the applied models simulate the compressor's behaviour based on characteristic curves provided by the manufacturer. For the thermal buffer storages all layers are resembled physically. Also heat exchange in between the layers is taken into account. The heat input is always fed into the appropriate layer. The thermal losses are credited to the building zone. The PV modules are realized by a two diode model that is suitable for mono- and polycrystalline SI as well as for CIS modules. The PV modules are connected to a MPP (maximum power point tracking) loop.

### 2.2.3. Central Hydrogen System for Seasonal Energy Storage

In order to increase the heating system self-sufficiency a seasonal energy storage system based on hydrogen is used to supplement the existing system. The hydrogen system consists of the following components:

- PEM Electrolyzer.
- Multi-stage (isothermal) hydrogen compressor.
- Hydrogen storage tank.
- PEM Fuel Cell.

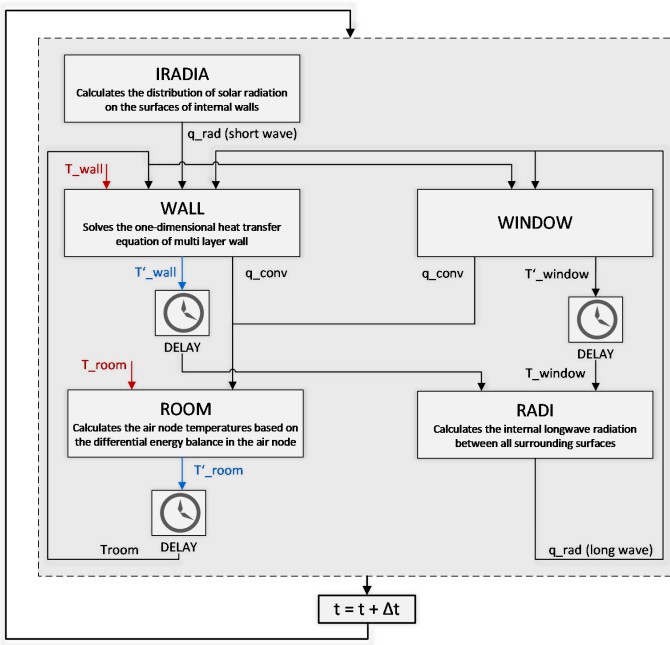

**Figure 2.** Building model node interconnections. Own representation based on [22].

In this system surplus electrical power of the building cluster is used to produce hydrogen via water electrolysis, which is afterwards compressed and stored at 35 MPa in a suitable cylinder. The PEM fuel cell is operated in a power-driven control mode to balance occurring power deficits. Additionally, the possibility of recovering the waste heat from the electrolyzer and the fuel cell, e.g., by feeding a district heating network, is regarded. All components and the system are modelled using the INSEL simulation environment.

The Hydrogen System is automatically rule-based dimensioned based on the electrical load profile of the building cluster by using a python script. The following rules are applied.

Electrolyzer

The rated power is defined by the maximal electrical power of the load profile. With the rated power the active cell area is defined (50 cm$^2$, 300 cm$^2$ or 1250 cm$^2$). Finally the number of needed cells can be calculated with the rated power, the active cell area and the assumed cell voltage at the maximal operating current density.

Compressor

The maximal hydrogen delivery rate is calculated based on the electrolyzer full load operation

$$\dot{m}_{H_2,max} = I_{max}/(zF) \cdot N \tag{1}$$

Based on the hydrogen production rate, the needed compressor power is estimated as described by Christensen [23].

Storage Cylinder

The size of the storage cylinder is calculated with the maximal seasonal surplus energy which could be stored from June to October.

$$E = \int_{June}^{October} P_{surplus} \cdot dt \tag{2}$$

With an assumed energy demand for the hydrogen production of $50\,\text{kWh}\,\text{kg}^{-1}$ and the targeted storage pressure, the needed cylinder volume is determined.

Fuel Cell

The rated power is defined by the minimal electrical power of the load profile (maximal power deficit of the building cluster). As for the elctrolyzer the cell area results from the rated power ($200\,\text{cm}^2$ or $400\,\text{cm}^2$), the number of cells from the cell voltage (0.63 V) and the maximal current density ($0.4\,\text{A}\,\text{cm}^{-2}$) with data from Bai et al. [24].

### 2.2.4. Small Wind Turbines

For the small wind power generation a characteristic curve model of an Aeolos-V 5 kW vertical wind turbine is used. The corresponding power output according to the wind speed can be found in Table 2. The installation height is assumed to be 15 m above ground and it is set, that for each building one small wind turbine is installed resulting in a total number of 10 small wind turbines. The measured wind data is scaled up to this height according to the wind profile power law that is resembled by the following equation:

$$V_2 = V_1 * \left(\frac{h_2}{h_1}\right)^{\alpha} \tag{3}$$

where $V_1$ [m/s] and $V_2$ [m/s] are the different wind speeds, $h_1$ [m] and $h_2$ [m] are the corresponding heights and $\alpha$ [-] is the roughness exponent of the environment. $\alpha$ is assumed to be 0.28 which corresponds to terrain with obstacles up to 15 m like forests, settlements, towns, etc. according to [25].

**Table 2.** Characteristic power output of an Aeolos-V 5 kW vertical small wind turbine.

| Wind Speed [m/s] | 1 | 2 | 3 | 4 | 5 | 6 | 7 | 8 | 9 | 10 | 11 | 12 | 13 | 14 |
|---|---|---|---|---|---|---|---|---|---|---|---|---|---|---|
| Power [kW] | 0 | 0.05 | 0.2 | 0.4 | 0.6 | 0.8 | 1.3 | 2.0 | 3.0 | 3.9 | 4.5 | 5.0 | 5.2 | 5.3 |

### 2.3. Household Electricity Demand

Regarding the household electricity demand, for four buildings measured data was available in a high resolution of 5 s and in sufficient quality to be used as model input. For the remaining six buildings, household electricity profiles were created corresponding to the real household size using the Loadprofile Generator tool [26] that is based on a bottom up model for electrical load generation. The household electricity demand of the previous same weekday is used for the prediction of household electricity demand.

### 2.4. Domestic Hot Water Demand

For the DHW demand, also from four buildings measured data was available in a resolution of 30 s and in sufficient quality to be used as model input. For the remaining six buildings, DHW profiles were created corresponding to the real household size using the Loadprofile Generator tool [26] that is also used for the generation of household electricity load profiles. The DHW demand of the previous same weekday is used for the prediction of DHW demand.

### 2.5. Building Model Calibration

In order to improve the model prediction accuracy, the combined decentral building and energy supply system models are calibrated on the basis of high-resolution measurements of the individual heat pumps electricity demand. Thereby the working steps are the following:

- Preprocessing of monitoring and weather data (filling of measurement gaps, unification of time steps).
- Application of a (genetic) optimization algorithm to the building parameters such as operating limit temperature, air exchange rate, room set point temperature, hysteresis of the buffer storage (heating and DHW) as well as the room heating hysteresis.
- The calibration takes place as INSEL-Python co-simulation using the DEAP (Distributed Evolutionary Algorithms in Python [27]) toolbox. The simulation is performed in one minute time steps, but the results and measured values are averaged to 60 min, otherwise a calibration is sometimes not possible due to the high fluctuation of the heat pump operation.
- As last step an assessment of the calibration results is carried out for a validation period that differs from the calibration period with a with a different data set.

The targetfunction is the normalized root mean square error (NRMSE) between measured and predicted values. The NRMSE is used instead of e.g., the mean absolute percentage error (MAPE), as this would be less conclusive in the case of high load peak fluctuations as they occur with heat pumps.

$$NRMSE = \frac{\sqrt{\frac{\sum_i^n (\hat{y}_i - y_i)^2}{n}}}{} * \frac{1}{y_{max} - y_{min}} \tag{4}$$

where $\hat{y}_i$ is the predicted value and $y_i$ is the measured value at the respective time step.

This is applied to the power consumption of the system, in this specific case the electrical demand of the heat pump. It should be noted, that for six of the buildings no measurement of the DHW demand were available due to incorrectly installed sensors. For those buildings, a DHW load profile corresponding to the real household size was generated using the previously mentioned Loadprofile Generator tool [26]. For the other buildings measurement data was parsed to a one minute resolution. The measured DHW demand of the previous same weekday is used for the validation and the prediction process.

### 2.6. Optimization of Scheduled Heat Pump Operation

As with the the model calibration, a genetic algorithm based on the DEAP toolbox in python is applied to each individual building and energy supply system model, that varies the heat pumps operational behaviour. This is also realized as INSEL-Python co-simulation. Thereby the heat pump operation is optimized for 24 h. The heat pump operation modes are the following:

- **Mode -1:** Normal operation.
- **Mode 0:** No heat pump operation. The optimization algorithm cannot select this mode directly, as this would cause the heat pump to go out of operation at every possibility, e.g., in a cost optimization scenario. This mode is used to realize possible blocking times by a sub script.
- **Mode 1:** Forced heat pump operation to load the space heating thermal buffer storage.
- **Mode 2:** Forced heat pump operation to load the DHW thermal buffer storage.
- **Mode 3:** Forced heating of the buildings thermal mass. Thereby the heat pump can operate reguarly and load the thermal buffer storages according to demand.

At each simulation start the model is parametrized partly with real measured values like the ambient temperature as the start temperature of the external part of the building envelope. A model settling time of 3 h is taken into account. The optimization is carried out for the following 24 h after the settling time in 5 min intervals. 5 min intervals are

chosen due to the fact, that longer intervals have shown (e.g., 20 min were tested) lesser potential as the heat pump's energy demand is less precisely scaleable. Also a time span of 5 min is assumed to be the minimum running time within this work as Green et al. have shown, that cycling times of less than 6 min reduce the heat pumps performance especial in the case of air source heat pumps but also for ground source heat pumps [28]. This is accompanied by the fact, that short cycling times of heat pump compressors can have negative effects on the heat pumps mechanical components lifetime (compressor motor, compressor and electrical contactors) [29]. The optimizations rebound effect is taken into account for the following 3 h after the optimization as this period is also included in the fitness evaluation of each result. The simulations run with an internal time step of 1 min. For the load curve that goes to the hydrogen system the time from 1 a.m. to 4 a.m. each day is not optimized. The reason for this is the process of creating a load profile over the entire year from the optimized daily profiles.

### 2.6.1. PV Self-Consumption

For the daily optimization case of PV self-consumption, the aim is to minimize the accumulated amount of electricity that is drawn from the grid. This helps to (i) reduce demand peaks, (ii) lowers the amount of PV electricity that is feed to the grid and (iii) poses the most economical solution for the building owner as today, the household electricity price in Germany is more than three times as high as the fixed feed in tariff for small (<10 kWp) newly installed PV systems. As such, for PV the fixed price feed in tariff for new systems installed in August 2021 would be 0.736 €/kWh [30] whereas the average household electricity price in Germany in 2021 is 0.33 €/kWh [31]. This results in the following target function for which the result is to be minimized:

$$E_{fromGridsum} = \int_{optstart+3h}^{optend} (E_{Household} + E_{Heatpump} - E_{PV}) \cdot dt \tag{5}$$

Whereas $E_{fromGridsum}$ [kWh] is the total amount of electrical energy take from the grid during the optimization time span, $E_{Household}$ [kWh] is the households electricity demand for each time step, $E_{Heatpump}$ [kWh] is the heat pumps electricity demand for each time step and $E_{PV}$ [kWh] is the PV systems electricity production for each time step. For the single building optimization this equation is directly applied to the model outputs. For the cluster optimization, first it is assumed that all electricity flows are balanced within the cluster.

### 2.6.2. PV and Wind Self-Consumption

For the daily optimization case of PV and wind self-consumption the same approach is used as it is described in Section 2.6.1. Only the additional wind production $E_{Wind}$ is added to the target function resulting in the following equation:

$$E_{fromGridsum} = \int_{optstart+3h}^{optend} (E_{Household} + E_{Heatpump} - E_{PV} - E_{Wind}) \cdot dt \tag{6}$$

### 2.7. Calculation of Carbon Dioxide Emission Savings and Prevention Costs

All measures to increase the self-consumption of the building cluster are compared based on its potential to reduce the annual carbon dioxide emissions of the cluster. Table 3 shows the emission factors for the power and heat source, based on available data for Germany.

The overall annual carbon dioxide emission saving is calculated as follows:

$$m_{CO_2,reduced} = (E_{fromGrid,base} - E_{fromGrid,variant}) \cdot e_{power} + E_{HeatToNetwork} \cdot e_{heat} \tag{7}$$

With the electrical energy needed from the grid in the base scenario $E_{fromGrid,base}$ [kWh] and the system variant $E_{fromGrid,variant}$ [kWh], the emission factor for electricity

$e_{power}$ [kg kWh$^{-1}$], the amount of recovered waste heat $E_{HeatToNetwork}$ [kWh] and the emission factor for district heating $e_{heat}$ [kg kWh$^{-1}$].

**Table 3.** Emission factors of power and heat.

| Source | Emission Factor | Source |
|---|---|---|
| Power From Grid | 0.408 kg kWh$^{-1}$ | [32] |
| District Heating | 0.243 kg kWh$^{-1}$ | [33] |

For the systems where additional investment costs are needed, a cost factor $c_{CO_2-saving}$ is calculated, which determines the overall costs per reduced mass of carbon dioxide emissions $m_{CO_2,reduced}$. All systems are calculated based on an assumed lifetime of $t_{life} = 20a$.

$$c_{CO_2,saving} = \frac{C_{invest} + C_{energy} \cdot t_{life}}{m_{CO_2,reduced} \cdot t_{life}} \tag{8}$$

In this case the energy cost consists of the mass of hydrogen which must be provided by external delivery, e.g., via trailer, to balance the deficit of hydrogen production within the building cluster (see Section 3.6). The specific cost for the calculation of $C_{invest}$ and the costs for external hydrogen deliveries are summarized in Table 4.

**Table 4.** Specific investment costs of the system components.

| Component | Specific Costs | Unit | Source |
|---|---|---|---|
| PEM Electrolyzer | $2000 + 1000 \cdot P_{EL}$ | €kW$^{-1}$ | [19] |
| Compressor | $2545 \cdot P_{comp}$ | €kW$^{-1}$ | [23] |
| Hydrogen Storage | $15 \cdot m_{H_2} \cdot LHV_{H_2}$ | €kWh$^{-1}$ | [19] |
| PEM Fuel Cell | $2000 + 1000 \cdot P_{EL}$ | €kW$^{-1}$ | [19] |
| Battery | $500 \cdot E_{bat}$ | €kWh$^{-1}$ | [34] |
| Small Wind Turbine (5 kW) | 24,000 | € | own assumption |
| Hydrogen Delivery | 9 | €kg$^{-1}$ | own assumption |

*2.8. Weather Data*

As simulation input the following measured weather data as well as prediction data are used: ambient temperature, global radiation and wind speed. For the measured data, the ambient temperature is measured on site with a resolution of 1 min. There also exists a measurement station for global radiation and wind speed on site. Unfortunately, this has been recently installed and therefor no measurements for a continuous year are available so far. Hence measurements from a nearby weather station situated close to the town of Ilshofen are used for those two parameters. Random samples indicate, that those global radiation measurements show little difference to available on-site measurements. The use of wind data that is not directly measured on side can be justified by the fact, that this study doesn't aim at a qualitative site potential assessment, but on assessing the load management potential of the cluster in general. The wind and global radiation data are collected with a 1 h time resolution. Wind measurements are taken at a height of 2.5 m. The prediction data (ambient temperature, global radiation with a 15 min resolution) comes from commercial provider for 72 h in advance and is updated multiple times per day. For this study for each day, the weather prediction that is delivered at 11:15 p.m. is taken as the basis.

**3. Results**

*3.1. Building Model Calibration Outcome*

The calibration of the building and decentral energy supply system models was carried out with measured demand and weather data for the time between 01.01.2020

and 31.03.2020 and the validation was carried out for the time between 01.04.2020 and 07.04.2020. For some buildings slightly other time spans were used due to gaps in the measured data. The initial model parameter and the calibration parameter range are presented in Table 5. The main results are shown in Tables 6 and 7. A visualization of the validation results for each model is shown in Figures A1 and A2.

**Table 5.** Initial model parameter and calibration parameter range.

| Building ID | Indoor Temp [°C] | Hyst Space [°C] | Hyst DHW [°C] | Hyst Room [°C] | Heating Stop Temp [°C] | Air Ex Rate [1/h] |
|---|---|---|---|---|---|---|
| Initial Param. | 20.0 | 5.0 | 5.0 | 2.0 | 10.0 | 0.30 |
| Param. Range | 15.0–25.0 | 0.01–10.0 | 0.01–10.0 | 0.01–3.0 | 8.0–18.0 | 0.00–0.60 |

**Table 6.** Model prediction error after calibration.

| Building ID | NRMSE Original | NRMSE Calibrated | E, Measured [kWh] | E, Predicted [kWh] |
|---|---|---|---|---|
| 01 | 30.8% | 18.4% | 105.1 | 100.0 |
| 02 | 40.6% | 26.1% | 81.1 | 72.4 |
| 09 | 49.8% | 47.6% | 92.4 | 92.6 |
| 10 | 68.3% | 32.9% | 62.8 | 62.0 |
| 12 | 32.2% | 29.7% | 138.1 | 133.4 |
| 19 | 41.7% | 32.7% | 117.1 | 103.1 |
| 20 | 19.0% | 19.0% | 83.6 | 83.0 |
| 22 | 45.4% | 23.8% | 53.2 | 46.8 |
| 24 | 45.8% | 28.3% | 68.7 | 56.1 |
| 25 | 36.5% | 20.8% | 61.7 | 57.2 |

**Table 7.** Model parameter after calibration.

| Building ID | Indoor Temp [°C] | Hyst Space [°C] | Hyst DHW [°C] | Hyst Room [°C] | Heating Stop Temp [°C] | Air Ex Rate [1/h] |
|---|---|---|---|---|---|---|
| 01 | 16.7 | 2.3 | 0.01 | 0.7 | 17.0 | 0.20 |
| 02 | 16.7 | 1.7 | 0.1 | 0.01 | 14.0 | 0.04 |
| 09 | 20.3 | 7.0 | 7.3 | 2.2 | 11.0 | 0.02 |
| 10 | 15.7 | 0.3 | 0.3 | 0.3 | 14.0 | 0.00 |
| 12 | 19.3 | 7.7 | 8.3 | 0.7 | 13.0 | 0.32 |
| 19 | 20.0 | 4.0 | 4.7 | 0.4 | 9.0 | 0.10 |
| 20 | 20.3 | 0.7 | 4.7 | 0.4 | 15.0 | 0.60 |
| 22 | 16.7 | 5.0 | 0.3 | 0.8 | 17.0 | 0.12 |
| 24 | 16.0 | 1.3 | 0.3 | 0.3 | 14.0 | 0.04 |
| 25 | 17.3 | 0.1 | 0.1 | 0.4 | 15.0 | 0.00 |

At average the NRMSE is reduced by 32% through the calibration process. It can be seen that for those buildings where measured DHW demand was available (building ID 20, 22, 24 and 25) the accuracy tends to be better. For most of the models the indoor set point temperature that results out of the calibration process is lower than a set point temperature of 20 °C that is generally assumed for residential building living areas. This can be credited to the fact, that normally only a part of the building is heated whereas the model set point temperature represents the average temperature through the entire building zone. In Table 7 the low space heating hysteresis occurs when the space heating buffer storage is in the rear flow of the heating circuit. The low air exchange rates indicate a ventilation system with heat recovery. The really low DHW hysteresis of building 01 and building 02 can be explained by the fact, that for those buildings no measured DHW

profiles were available and thus the algorithm tends to mitigate this negative effect by trying to reduce DHW demand fluctuations by lowering the specific time span when the heat pump is activated to the minimum.

### 3.2. Impact of Weather Prediction Inaccuracy

#### 3.2.1. Building Heat Demand

The effect of weather prediction inaccuracy on the heating demand prediction quality was investigated for a single building out of the cluster. Therefor the same simulation was carried out with measured weather data, with forecasted weather data for the same time period with a forecast horizon of 0–24 h and with a forecast horizon of 24–48 h. In Table 8, the NRMSE is presented in between the simulation results and the buildings measured demand. A visualization of the resulting power demand curves is shown in Figure A3 and a visualization of a comparison of the accumulated electricity demand is shown in Figure A4. It can be seen, that the NRMSE increases by 17.6% when relying on forecast data for 0–24 h in advance instead of measured weather data. It increases by 45.0% when relying on forecast data for 24–48 h in advance. It can be said, that for the buildings heating demand, model inaccuracy has a larger impact than weather forecast uncertainties. In contrary the results for the cumulative electricity demand are good and show little influence of the prediction inaccuracy. This leads to the assumption that the highest uncertainties com from the timing of the heat pump system and thus from control of the real system and user behaviour.

**Table 8.** Comparison of heating electricity demand prediction error according to weather data forecasting duration.

|  | Measured Weather Data | 24 h Forecast | 48 h Forecast |
|---|---|---|---|
| NRMSE | 14.2% | 16.7% | 20.6% |

#### 3.2.2. PV Power Generation

A similar evaluation as for the electrical heating demand prediction error was carried out for the PV generation prediction error over a period of 6 months. In Table 9 it can be seen, that the NRMSE is 9.7% when relying on forecast data for 0–24 h in advance instead of measured weather data. Interestingly it stays approx. the same when relying on forecast data for 24–48 h in advance. Only when relying on forecast data for 48–72 h in advance it increases by 10.5%.

**Table 9.** Comparison of PV generation prediction error according to weather data forecasting duration.

|  | 24 h Forecast | 48 h Forecast | 72 h Forecast |
|---|---|---|---|
| NRMSE | 9.7% | 9.6% | 10.6% |

### 3.3. Best Parameters for Schedule Optimization

The best parameters for the genetic algorithm were determined iterative for the optimization of the heat pump operation (creation of 24 h schedules) with regard to the self-consumption of PV electricity. The number of generations was varied step wise between 50 and 300 and the population size was varied step wise between 5 and 70 individuals. The crossover and mutation probability was varied in each case in 0.1 steps from $p_c/p_m = 0$ to $p_c/p_m = 1.0$. Ten computational runs were performed for each parameter combination of the algorithm to reduce the effects of randomness, which occur primarily when the number of generations and the population size are small. Thus, 72,600 different parameter combinations were examined. The number of ten computational runs was considered as a compromise between accuracy and computational demand. E.g., for the case of a low number of generations of 50 and a low population size of 5 ($p_c = 0.9$ and $p_m = 0.3$)

the standard deviation would be 8.2% for 5 computational runs and 4.1% in the case of 10 computational runs.

The overall result is shown in Figure 3a,b. The percentages indicate the improvement in terms of cost savings resulting from avoided electricity consumption. It can be seen that the number of generations has low influence. This can be contributed to the fact that the selected minimum number of 50 already represents a sufficiently large number. For the population size, the results approach an optimum with increasing number of individuals. The relatively small improvement of up to 28% is due to the fact that this value represents the mean value from the various crossover and mutation probabilities and thus many bad combination possibilities are included. Looking in detail at the results of the varied crossover and mutation probabilities in Figure 3b, the best results are shown for a crossover probability of $0.7 \leq p_c \leq 1.0$ and for a mutation probability of $0.1 \leq p_m \leq 0.3$. Based on these results the crossover probability is set to $p_c = 0.9$ and the mutation probability is set to $p_m = 0.3$ with the specific results shown in Figure 4b, and the number of generations and the population size are set to 50 with the specific results shown in Figure 4a. These parameters present a good compromise between the quality of the result and the required computing time (2500 simulation runs, 30 min). Due to the forecast inaccuracy, it is also not necessary to determine the absolute optimum, since it can be assumed that a prediction inaccuracy will always occur in real operation. By optimizing the prediction model and the parameters of the GA, the computing time for the optimization is reduced from about 150 min to 30 min.

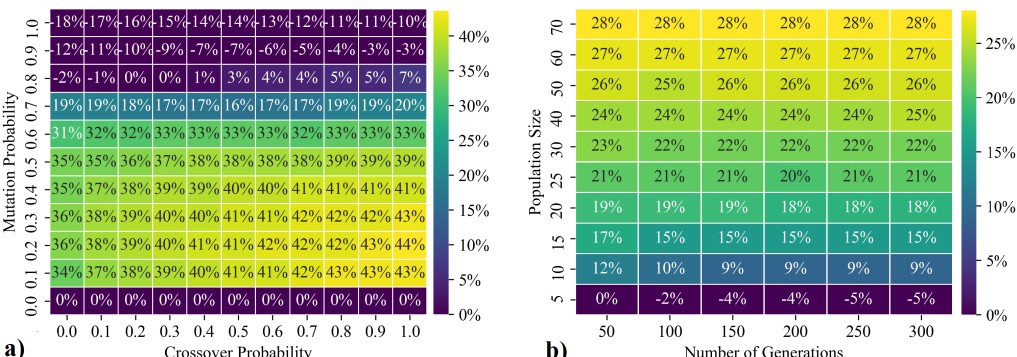

**Figure 3.** Parameter variation genetic algorithm without constrains. (**a**): Dependency on mutation and crossover probabilities for all population sizes and number of generations; (**b**): Dependency on population size and number of generations for all mutation and crossover probabilities.

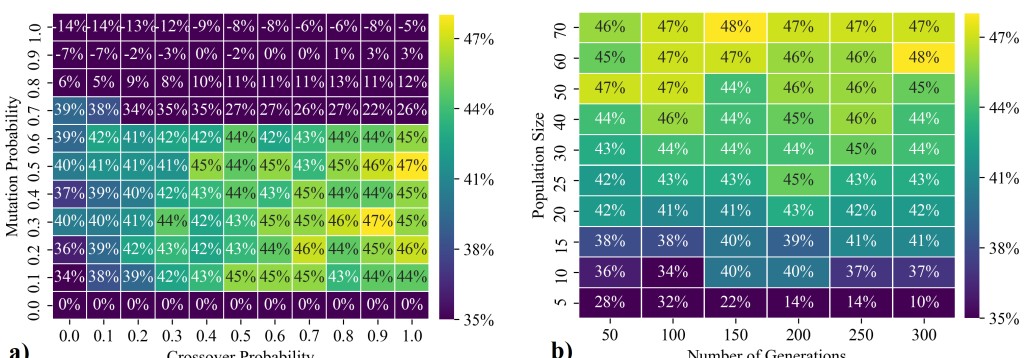

**Figure 4.** Parameter variation genetic algorithm with constrains. (**a**): Number of generations and population size set to 50; (**b**): Crossover probability set to $p_c = 0.9$ and mutation probability set to $p_m = 0.3$.

### 3.4. Single Building Operation

Optimized Self-Consumption

As starting point the PV-self-consumption of a single building was optimized for one year on a daily basis. The considered building has a heat pump with approx 3.5 kW electric power consumption and a rather large PV system with 28.8 kWp. The buffer storages are defined in the model to be 750 L for DHW and 1500 L for space heating. These parameters can be considered as ideal conditions. An exemplary daily operation without optimization can be seen in Figure 5 whereas the results with optimization are shown in Figure 6. It can be seen that the algorithm shifts the operation of the heat pump to the period with sufficient PV power generation reducing the electricity demand from the grid during the evening hours and night time.

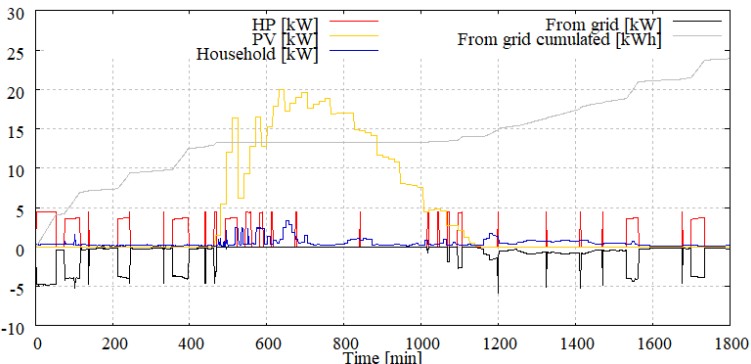

**Figure 5.** Single building daily electrical demand without optimization.

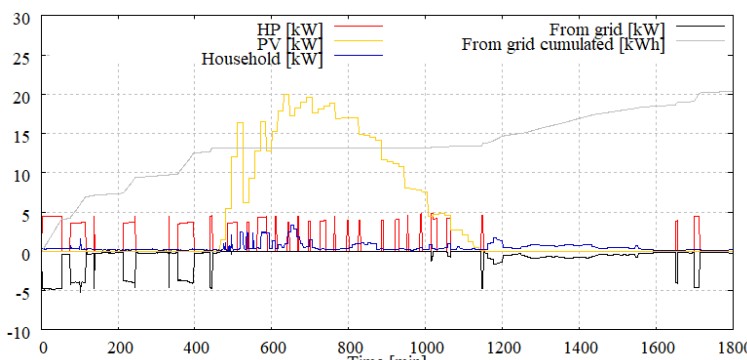

**Figure 6.** Single building daily electrical demand with optimization.

Table 10 shows the optimization results on a monthly basis. Thereby Opt is the assumed reduction of electricity demand from the grid based on predicted weather data whereas Meas Opt is the resulting demand reduction which would have occurred with the real weather situation and Ad Hoc is the same result with the assumption of an ad hoc algorithm that would switch to normal operation instead of the schedule in case the prediction inaccuracy would lead to a higher demand. A decline between Opt (annual average of 9.8%) and Meas Opt (annual average of 3.0%) can be clearly recognized resulting from the prediction inaccuracy whereas 2/3 can be contributed to weather forecast inaccuracy and 1/3 results from the inaccuracy of household electricity and DHW demand prediction. With the inclusion of an ad hoc algorithm that would rule out schedules that worsen the result during the daily operation, an annual improvement of 4.9% could be achieved. In total 221.2 kWh (234.6 kWh with an ad hoc algorithm) of lesser electricity would have been demanded from the grid. The improvement on a monthly basis in general is not very high reaching at maximum 6.6% in April (8.1% in May with ad hoc algorithm). However on the best days, an theoretical improvement of Opt = 44% and an improvement with measured weather data of Meas Opt = 33% could be achieved.

**Table 10.** Monthly PV self-consumption improvement for a single building.

|          | Jan  | Feb  | Mar   | Apr   | May   | Jun   | Jul   | Aug   | Sep   | Oct  | Nov  | Dec  |
|----------|------|------|-------|-------|-------|-------|-------|-------|-------|------|------|------|
| Opt      | 3.8% | 7.0% | 10.4% | 10.6% | 14.9% | 15.4% | 16.6% | 12.5% | 10.8% | 6.7% | 4.5% | 3.9% |
| Meas Opt | 0.9% | 4.6% | 4.0%  | 6.6%  | 5.0%  | 1.6%  | 0.6%  | 3.1%  | 3.3%  | 3.9% | 1.4% | 1.1% |
| Ad Hoc   | 1.2% | 5.2% | 4.8%  | 8.0%  | 8.1%  | 5.0%  | 5.9%  | 6.8%  | 5.8%  | 4.7% | 1.7% | 1.5% |

If we take a look at the results on a daily basis in Figure 7, it can be seen that within this month under good conditions improvements of up to 23% can be achieved. These are evened out by those days with less or no optimization potential. There can also be seen two days with a significant decrease of up to −15%. The general daily fluctuation can be explained the fact that the optimization potential varies according to PV production and demand by DHW and space heating. The discrepancy in between Opt and Meas Opt is to be credited to prediction inaccuracies. E.g., the worst result during April occurred on April 22. During that day the prediction inaccuracy for DHW and household electricity demand was unusually high. This happened possibly due to the fact that this day was Easter Monday in 2019 which is a holiday in Germany and therefor the user behaviour was not like it should have been on a work day. The 23% Meas Opt result on April 11. that was more than double of the predicted Opt result happened because on that specific day the measured amount of global radiation was 38% higher compared to the predicted amount.

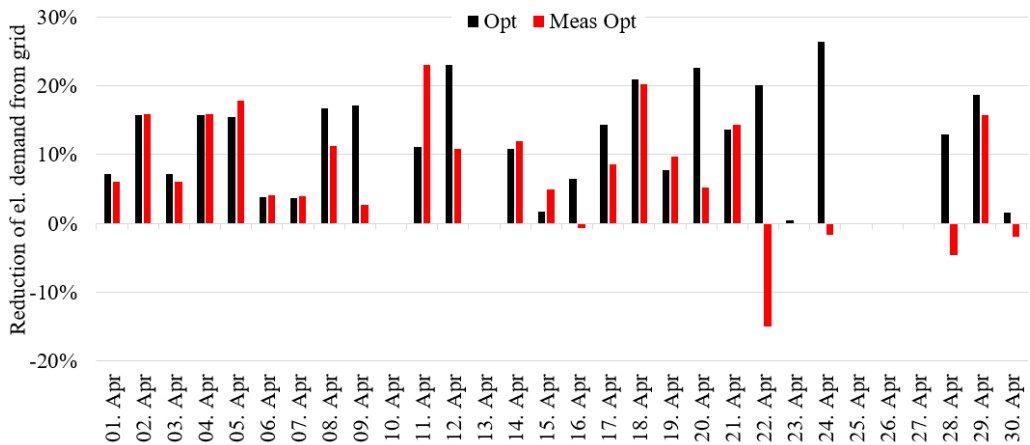

**Figure 7.** Daily optimization results for a single building for the month April.

In general the results without ad hoc algorithm are worse during summer where DHW prediction inaccuracy has a higher impact due to the lack of space heating demand. In opposite, the theoretical optimization potential during summer would be up to Opt = 16.6% on a monthly basis.

In a next step the thermal buffer storage size was doubled. The monthly results are shown in Table 11. Thereby mainly the increase of the DHW storage size has an effect during the summer months. The average annual predicted improvement is Opt = 12.8% whereas the average improvement including the prediction inaccuracy is Meas Opt = 3.7%. With the inclusion of an ad hoc algorithm Ad Hoc = 5.6% could be achieved on annual average. This is only slightly better than with the previous thermal buffer storage sizes. In total 257.0 kWh (347.0 kWh with an ad hoc algorithm) of lesser electricity would have been demanded from the grid.

**Table 11.** Monthly PV self-consumption improvement for a single building with doubled storage size.

|          | Jan  | Feb  | Mar   | Apr   | May   | Jun   | Jul   | Aug   | Sep   | Oct   | Nov  | Dec  |
|----------|------|------|-------|-------|-------|-------|-------|-------|-------|-------|------|------|
| Opt      | 4.7% | 9.8% | 12.5% | 15.0% | 20.9% | 18.4% | 19.5% | 17.0% | 15.9% | 10.0% | 5.6% | 4.4% |
| Meas Opt | 1.1% | 5.8% | 3.0%  | 4.9%  | 2.2%  | 7.0%  | 2.6%  | 5.9%  | 5.4%  | 2.9%  | 2.1% | 1.7% |
| Ad Hoc   | 1.8% | 6.0% | 4.1%  | 6.9%  | 5.7%  | 9.4%  | 8.5%  | 8.8%  | 7.0%  | 4.9%  | 2.7% | 2.1% |

### 3.5. Building Cluster Operation

3.5.1. Optimized PV Self-Consumption

Regarding cluster operation the PV self-consumption of a cluster of 10 buildings was optimized in the same way as on single building level. Thereby solely through the interconnection of the buildings individual electricity demand and generation the electricity demand from the grid would be reduced by 10% from 54,436 kWh to 49,140 kWh as well as the grid infeed would be reduced by 6% from 84,473 kWh to 79,176 kWh for the year 2019.

An exemplary daily operation without optimization is shown in Figure 8 whereas the results with optimization are shown in Figure 9. In these figures the model settling time, that is not taken into account for the optimization can be seen for the first 110 min. To evaluate the results the accumulation of the electricity demand starts after 180 min (grey line). The monthly results can be found in Table 12.

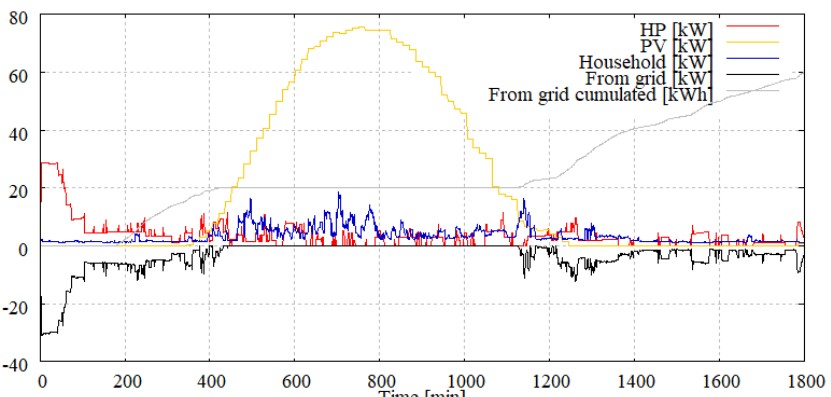

**Figure 8.** Cluster daily electrical demand without optimization.

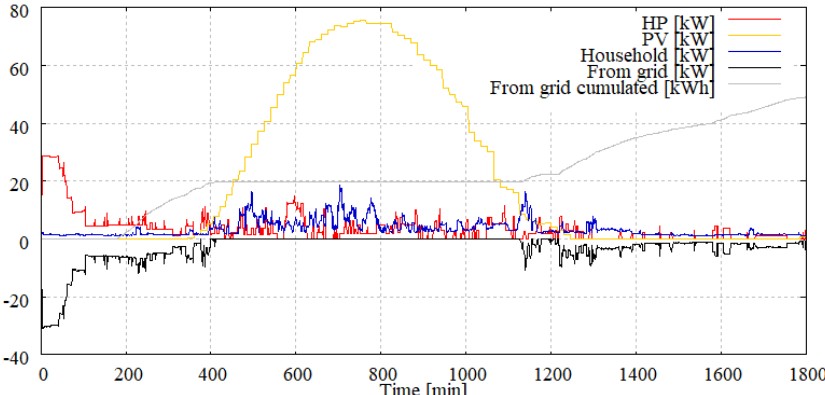

**Figure 9.** Cluster electrical demand with optimization.

**Table 12.** Monthly PV self-consumption improvement for the building cluster.

|          | Jan   | Feb    | Mar   | Apr    | May   | Jun   | Jul    | Aug   | Sep   | Oct   | Nov    | Dec    |
|----------|-------|--------|-------|--------|-------|-------|--------|-------|-------|-------|--------|--------|
| Opt      | 1.6%  | 3.3%   | 3.5%  | 5.0%   | 6.2%  | 6.6%  | 4.3%   | 4.5%  | 5.0%  | 4.3%  | 2.4%   | 1.5%   |
| Meas Opt | 0.0%  | −0.1%  | 0.1%  | −0.2%  | 0.2%  | 0.4%  | −0.4%  | 0.8%  | 0.3%  | 0.0%  | −0.4%  | −0.1%  |
| Ad Hoc   | 0.3%  | 0.5%   | 0.9%  | 0.6%   | 1.1%  | 1.4%  | 0.6%   | 1.2%  | 1.2%  | 1.1%  | 0.5%   | 0.3%   |

The average predicted annual improvement is Opt = 4.1% whereas the average annual improvement including the prediction inaccuracy is neglectable at Meas Opt = 0.5%. Even with the addition of an ad hoc algorithm the average annual improvement would just increase to Ad Hoc = 0.8%. On the best days, an theoretical improvement of Opt = 20% and an improvement with measured weather data of Meas Opt = 15% could be achieved. Compared to the single building optimization the results are considerably worse. This can be contributed to the to the fact, that surplus electricity is also used within the cluster before the optimization. Also the ratio of installed PV system size and heat pump electrical demand for some of the buildings within the cluster is unfavorable contributing to these results. In addition days with bad household electricity and DHW demand prediction in some buildings can't be ruled out accurately by an ad hoc algorithm due to the fact that, all demands are accumulated before the evaluation. The results on a daily basis for the month April in Figure 10 underline this outcome. Only on a few days would an improvement be possible.

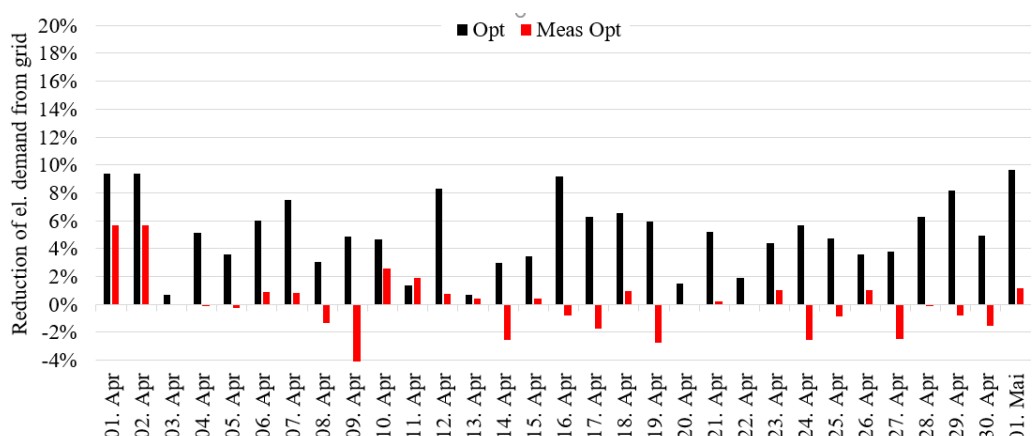

**Figure 10.** Daily optimization improvement for a building cluster with normal storage size for the month April.

In a next step the thermal buffer storage size was doubled. The monthly results can be found in Table 13. Thereby the effect ist minor. The average annual predicted improvement is Opt = 4.6% whereas the average annual improvement including the prediction inaccuracy is Meas Opt = 0.7%. With the addition of an ad hoc algorithm the average annual improvement would increase to Ad Hoc = 1.2%.

**Table 13.** Monthly PV self-consumption improvement for a building cluster with doubled storage size.

|          | Jan    | Feb   | Mar   | Apr   | May   | Jun   | Jul   | Aug   | Sep   | Oct   | Nov   | Dec   |
|----------|--------|-------|-------|-------|-------|-------|-------|-------|-------|-------|-------|-------|
| Opt      | 1.9%   | 4.1%  | 4.5%  | 5.3%  | 7.7%  | 5.4%  | 4.2%  | 5.4%  | 3.9%  | 4.5%  | 3.2%  | 1.8%  |
| Meas Opt | −0.2%  | 1.6%  | 0.7%  | 1.3%  | 1.0%  | 1.0%  | 0.5%  | 1.1%  | 0.4%  | 0.1%  | 0.5%  | 0.1%  |
| Ad Hoc   | 0.3%   | 1.9%  | 1.5%  | 1.5%  | 2.0%  | 1.7%  | 0.9%  | 1.7%  | 0.8%  | 1.0%  | 0.7%  | 0.4%  |

### 3.5.2. Optimized PV and Wind Self-Consumption

As a hypothetical approach the optimization was also applied to optimize the PV and wind self-consumption. This was done to show the theoretical potential as no wind speed prediction uncertainties were included. The monthly results can be found in Table 14.

**Table 14.** Monthly PV and Wind self-consumption improvement for a building cluster.

|          | Jan   | Feb  | Mar   | Apr  | May   | Jun   | Jul   | Aug   | Sep  | Oct   | Nov   | Dec  |
|----------|-------|------|-------|------|-------|-------|-------|-------|------|-------|-------|------|
| Opt      | 3.6%  | 7.4% | 5.6%  | 7.8% | 6.8%  | 6.6%  | 6.5%  | 5.3%  | 7.3% | 6.0%  | 5.3%  | 0.0% |
| Meas Opt | −0.2% | 2.2% | −0.6% | 0.5% | −0.3% | −0.4% | −0.1% | −0.2% | 1.1% | −0.3% | −1.9% | 0.1% |
| Ad Hoc   | 0.4%  | 2.8% | 0.9%  | 1.3% | 1.4%  | 1.3%  | 0.8%  | 1.3%  | 1.6% | 0.9%  | 0.9%  | 0.4% |

It can be seen that compared to the PV self-consumption optimization the results are slightly better in winter due to the additional wind production. The results on a daily basis for the month February are shown in Figure 11. The month February was chosen because the wind yield tends to be higher during winter months. This optimization on cluster level that includes the wind production suffers from the same problems as PV self-consumption optimization at cluster level. It is conceivable that a better optimization potential would be available at sites with better wind yields. Also results would be better if focused on single building level. This will be examined as a continuation of this work.

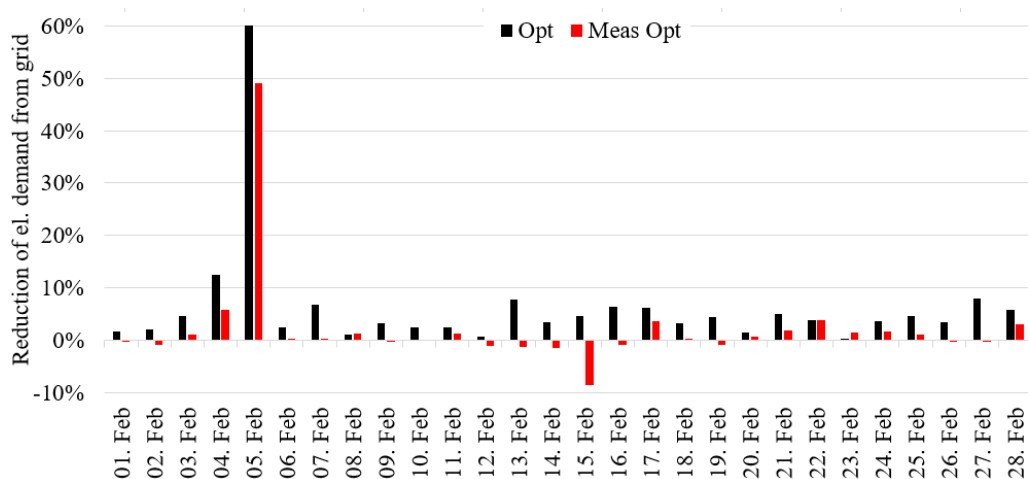

**Figure 11.** Daily optimization improvement (PV and wind) for a building cluster with normal storage size for the month February.

### 3.6. Adding of Hydrogen Systems with and without Small Wind Turbines

All in all the addition of three different measures to decrease the electrical power of the building cluster received from the power grid are compared and evaluated.

- Base Scenario: Decentral PV and Heat Pumps.
- Optimization: Optimized Heat Pump operation to increase PV self-consumption based on weather forecast data.
- Optimization + Hydrogen System: Seasonal hydrogen storage system with electricity-driven fuel cell to operate the heat pumps with surplus PV power stored from the summer months. Heat recovery from the electrolyzer and the fuel cell and heat injection into a district heating network are considered.
- Optimization + Hydrogen System + Small Wind Turbines (SWT): Additional small wind turbines to increase the power production throughout the year.

In this section the results of adding a hydrogen system and small wind turbines to the building cluster will be discussed. The resulting system parameters for both the hydrogen system and the hydrogen system with additional small wind turbines are shown in Table 15.

The adding of SWTs increases the rated power of the electrolyzer as more surplus power is available. As a result the hydrogen compressor as well as the storage tank volume must be up-scaled to match the increased hydrogen production. On the other hand, the installed fuel cell power decreases slightly, as the power deficit is damped through the wind power yield. The initial filling level of the storage tank must be lower than 100% for the system with wind turbines as hydrogen overproduction is achieved (see Figure 12).

Table 16 shows the annual amount of energy from the grid for different system combinations. Adding a hydrogen system with the described properties, equally with or without SWTs, basically leads to full autarky of the building cluster related to grid electricity.

However, an investment must be made. Table 17 shows the estimated system capital expenditures (CAPEX) of both systems. From the annual costs of hydrogen refills it is apparent that the hydrogen system alone is not self-sufficient related to the needed hydrogen (see Figure 13). Yearly cost of 4.724 € for hydrogen deliveries occur. The adding of small wind turbines on the other hand leads to hydrogen surpluses (see Figure 12) and allows an export of hydrogen with negative annual hydrogen costs. The specific costs for the saving of carbon dioxide emissions $c_{CO_2,saving}$ are shown in Figure 14. Under the assumed conditions the hydrogen system with heat recovery has the lowest specific costs of 1.23 € per saved kg of $CO_2$ emissions. The addition of small wind turbines does not decrease the specific costs, although hydrogen surpluses are achieved.

**Table 15.** System parameter overview.

| Parameter | Hydrogen System | SWT and Hydrogen System |
|---|---|---|
| Electrolyzer Power | 78 kW | 107 kW |
| Electrolyzer Cells | 45 | 62 |
| ELectrolyzer Cell Area | 300 cm$^2$ | 300 cm$^2$ |
| Max. Hydrogen prod. Rate | 1.51 kg h$^{-1}$ | 2.08 kg h$^{-1}$ |
| Storage Volume | 29.5 m$^3$ | 37.43 m$^3$ |
| Initial Storage Tank Filling | 35 MPa | 15 MPa |
| FC Power | 37.5 kW | 37.4 kW |
| FC Cells | 345 | 344 |
| FC Area | 400 cm$^2$ | 400 cm$^2$ |
| Compressor Power | 17.75 kW | 24.5 kW |

**Table 16.** Annual electrical energy from grid for different system variants.

| Parameter | w/o SWT | With SWT |
|---|---|---|
| $E_{grid}$ w/o $H_2$ | 48,385 kWh | 36,974 kWh |
| $E_{grid}$ w/ $H_2$ | 59 kWh | 70 kWh |

**Table 17.** Economic parameter overview.

| Parameter | Hydrogen System | SWT and Hydrogen System |
|---|---|---|
| Estimated System CAPEX | 597.852 € | 930.342 € |
| Annual Hydrogen Refill Costs | 4.724 € | −2.452 € |
| $c_{CO_2,saving}$ | 1.76 € kg$^{-1}$ | 2.23 € kg$^{-1}$ |
| $c_{CO_2,saving}$ (Heat Recovery) | 1.23 € kg$^{-1}$ | 1.55 € kg$^{-1}$ |

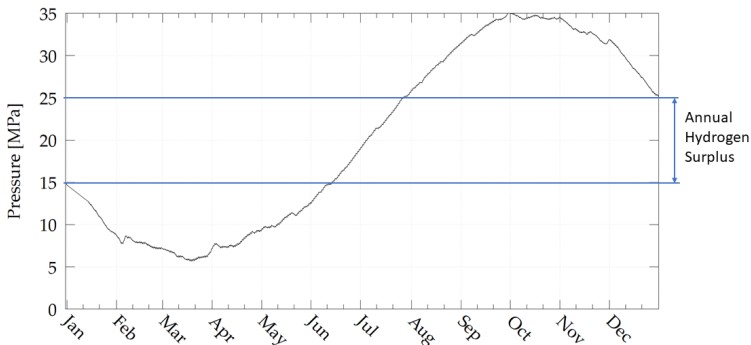

**Figure 12.** Annual progression of the hydrogen storage pressure with SWTs.

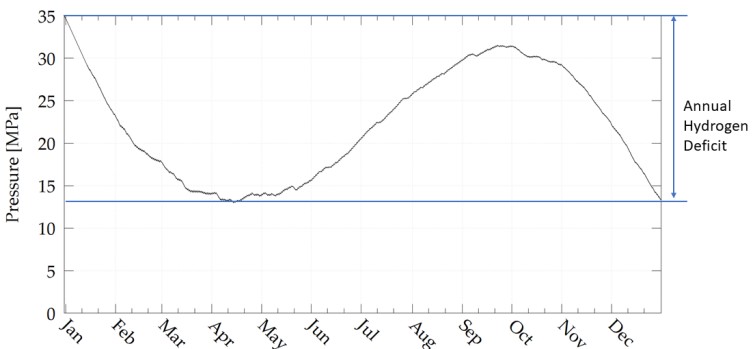

**Figure 13.** Annual progression of the hydrogen storage pressure.

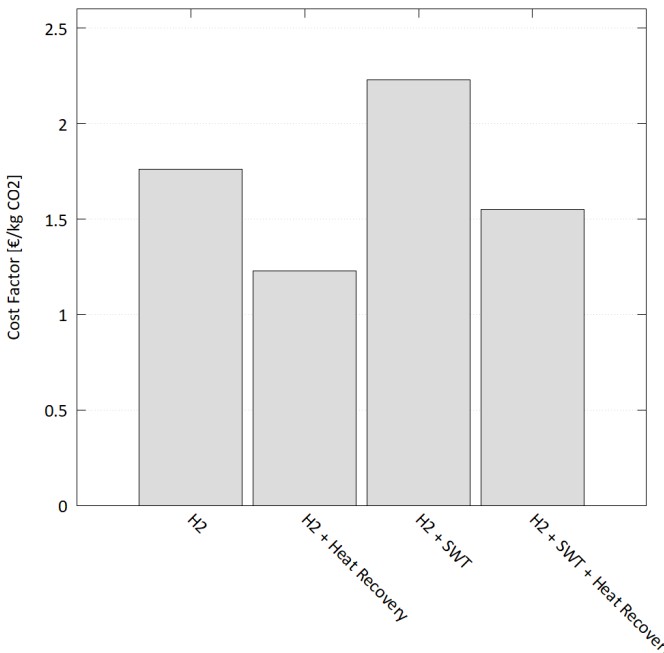

**Figure 14.** Specific costs of carbon dioxide savings.

## 4. Discussion

Within this work, the proposed modelling and optimization methodology was fully implemented and applied to a case study to evaluate different optimization goals such as an improvement of generated electricity self-consumption for decentral PV systems on a single building and cluster level as well as for small wind turbines combined with PV on a cluster level. To achieve a higher level of autarky and self-consumption in addition to the

optimization, the inclusion of a hydrogen system was proposed. Thereby the the following results were archived for the work packages modelling, optimization case studies and hydrogen system potential:

### 4.1. Modelling

Simulation models of the buildings and their energy supply systems were created for 10 residential buildings as well as calibrated and validated with the heat pumps measured electrical demand data. Thereby through the calibration an improvement of model accuracy, based on an assessment by NRMSE, of 32% was achieved. For the heating demand, the impact of weather prediction inaccuracy was 17.6% when relying on forecast data for 0–24 h in advance instead of measured weather. The NRMSE in that case was 16.7%. For PV generation the NRMSE for forecast data for 0–24 h in advance was 9.7%.

Furthermore the optimal GA parameters for the optimization of heat pump schedules were determined. This resulted in an optimal crossover probability of 0.9, an optimal mutation probability of 0.3 and an optimal number of generations and population size of 50, resulting in 2500 simulation runs per optimization. Through that and through a simplification of the prediction model details, the computational demand was reduced by 80%.

### 4.2. Optimization Case Studies

On single building level better optimization results were achieved compared to the cluster level. Still the optimization was often prone to solutions that show a negative result. This could be addressed by an ad hoc algorithm to reduce losses for days with bad prediction accuracy.

Overall the results showed a predicted annual average improvement of 9.8% and an improvement of 3.0% when considering the weather situation that actually occurred. With the addition of an ad hoc algorithm an annual average improvement of 4.9% could be archived. This worsening can be contributed to 2/3 to weather forecast inaccuracy and to 1/3 it results from the inaccuracy of household electricity and DHW demand prediction. On the best days an increase of PV self-consumption of up to 33% (44% without any prediction inaccuracies) could have been achievable. A further increase of thermal buffer storage sizes (doubling the size) only showed a marginal effect.

On cluster level the demand from grid and the infeed is already reduced by cluster interconnection of the buildings by 10% (demand) and 6% (infeed). Overall the cluster optimization results showed a predicted annual average improvement of 4.1% and a neglectable improvement of 0.5% when considering the weather situation that actually occurred (0.8% with the addiction of an ad hoc algorithm). Those small improvements can be contributed to the to the fact, that the surplus electricity generated by the PV systems is already partly used within the cluster before the optimization. Also the other buildings tend to have a more unfavorable ratio of installed PV system size compared to the heat pumps electrical demand contributing to a lower optimization potential. In addition, days with bad household electricity and DHW demand prediction in some buildings can't be ruled out accurately due to the fact that all demands are accumulated before the evaluation. On daily basis an increase of up to 15% in PV self-consumption was possible (20% without any prediction inaccuracies). The cluster operation also tends to have more days without optimization potential due to these reasons. For the cluster optimization a further increase of thermal buffer storage sizes (doubling the size) also only showed a marginal effect. On cluster level, the inclusion of wind generation showed similar results as for the sole PV self-consumption cluster operation. Here the inclusion of wind generation was a very hypothetical assumption as for the wind generation no wind speed prediction inaccuracy was included due to the fact that it is very complex and error prone compared to temperature and global radiation predictions. This could be counteracted by a model predictive control (MPC) setting that has the model run in a loop in parallel to the normal operation what would enable near real time optimizations.

In general increasing the size of the buffer storages only helped up to a certain size where in contrary this leads to the drawback of higher thermal losses, an increase of investment costs and more required space. The results showed that the increase of the DHW buffer storage on single building level had the most impact.

*4.3. Hydrogen System Potential*

The application of a hydrogen system for seasonal energy storage basically leads to full autarky, if the electrolyzer is designed to match the peak surplus power of the cluster and the fuel cell to match the peak power deficit of the building cluster. However, this measure comes with high investment costs and hence high carbon dioxide prevention costs ($1.76 \in kg^{-1}$). If the heat produced by the electrolyzer and the fuel cell is recovered and fed into a district heating network, the $CO_2$ prevention costs could be reduced by 30%. In our investigated scenario, the hydrogen system would depend on external hydrogen deliveries to maintain its operation in the winter months. The adding of small wind turbines to increase the power output of the cluster on the other hand leads to an annual surplus hydrogen production of the system, which could be sold. Nevertheless, the adding of small wind turbines results in even higher $CO_2$ prevention costs ($2.23 \in kg^{-1}$). The next steps aim at finding an optimum for the dimensioning of the hydrogen system based on the achievable $CO_2$ prevention costs.

## 5. Conclusions

In general it can be said, that the results observed in this work are in line with the experiences made in other publications, even if the specific application differs from them. As exambple, Zhou et al. [10] aimed to reduce reduce the peak load of an office buildings' air conditioning system by optimization and achieved an improvement of 33% in power costs with a computation time of 1–1.5 h. By Xu et al. [11] a variation of temperature set points was carried out to achieve optimal pre-cooling of the building during the following day resulting in a 21.3% reduction of energy costs. Kajl et al. [12] optimized the set points of a LEM system in a building on an university campus for two summer months, resulting in a 16% reduction of energy demand with a computation time of 30 min. In comparison within this work for the single building PV self-consumption optimization on the best days an improvement of up to 33% could have been archived that would have resulted in a cost saving of approx. 26%. For this the computation time was approx. 30 min.

Overall the proposed methodology was implemented successfully. The results have shown that the application of a hydrogen system for seasonal energy storage would allow full autarky but with high carbon dioxide prevention costs. Those could be reduced by up to 30% if the heat produced by the electrolyzer and the fuel cell would be recovered and fed into a district heating network.

Regarding the optimization scenarios, good results for the optimization on single building level were achieved on a daily basis ranging up at average to reductions of electricity demand from grid of over 16% in the summer months. However prediction inaccuracies tend to worsen the results significantly. Unfortunately at average over an entire year and especially for the cluster optimization better results would have been expected. This can be credited to several reasons. First, for many days the potential that relies on the ratio between heat demand and electricity production was minor with the installed system sizes. Second prediction errors for demand (household electricity and DHW) and weather (ambient temperature and global radiation) minored the accuracy of the optimization results and thus their impact to improve heat pump operation. In that case the biggest improvement could be achieved by applying more sophisticated prediction methods for household electricity and DHW demand. In general it can be said, that self-consumption optimization is more useful on single building level. Therefor the optimization should be carried out on single building level and the excess electricity should be provided to fulfil the other buildings demand afterwards. On cluster level it is more useful for applications like an aggregated operation to provide DR, e.g., to follow an aggregator's traces. Also for

real life applications the economics wouldn't be that favorable under today's regulatory conditions. As an example, for the single building optimization presented in this research, annual savings of 56 € would have been possible. A further limiting factor within this work is the fact, that so far uncertainties regarding the heat pumps availability and their control under real operational conditions were not included in the model approach. This might lessen the potential within real operation. To verify this an implementation to control heat pumps within the pilot site under real operational conditions is planned. It is also planned to further improve the optimization potential by a better DHW and household electricity demand prediction, e.g., through self-learning algorithms by ANN or k-nearest neighbors (kNN) algorithms. Also better over all annual results for the optimization could be possible in other climate zones with lesser seasonal difference as well as for sites with a higher wind yield. A dynamic inclusions of the hydrogen system within the optimization process could also be a promising option as well as the appliance of the developed methodology to optimize the charge and discharge of battery storages as well as for bi-directional charging of EV's.

**Author Contributions:** Conceptualization, M.B., D.L. and D.P.; methodology, M.B. and D.L.; software, M.B. and D.L.; validation, M.B. and D.L.; formal analysis, M.B. and D.L.; investigation, M.B. and D.L.; resources, M.B. and D.L.; data curation, M.B.; writing—original draft preparation, M.B. and D.L.; writing—review and editing, M.B. and D.P.; visualization, M.B. and D.L.; supervision, D.P. and D.S.; project administration, D.P. and D.S.; funding acquisition, D.P. and D.S. All authors have read and agreed to the published version of the manuscript.

**Funding:** This work emanated from research that was conducted with the financial support of the European Commission through the H2020 project Sim4Blocks, grant agreement 695965, from research within the Project "NEQModPlus", grant number 03ET1618B, funded by the German Federal Ministry of Economic Affairs and Energy (BMWi) and from research conducted within the project EnVisaGe Plus, grant number 03ET1465B, funded by the German Federal Ministry of Economic Affairs and Energy (BMWi).

**Institutional Review Board Statement:** Not applicable.

**Informed Consent Statement:** Not applicable.

**Data Availability Statement:** Data available on request due to restrictions (data privacy of building specific demand data).

**Conflicts of Interest:** The authors declare no conflict of interest.

## Abbreviations

The following abbreviations are used in this manuscript:

| | |
|---|---|
| $\alpha$ | Roughness exponent [-] |
| *ACO* | Ant colony optimization |
| *ANN* | Artificial neural network |
| *BFGS* | Broyden-Fletcher-Goldfarb-Shanno |
| *C* | Cost [€] |
| *c* | Cost factor [-] |
| *CAPEX* | Capital expenditures |
| *CIS* | Copper indium gallium selenide solar cell |
| *DEAP* | Distributed evolutionary algorithms in python |
| *DHW* | Domestic hot water |

| | |
|---|---|
| $e$ | $CO_2$ emission factor [kg/kWh] |
| $E$ | Energy [kWh] |
| $EA$ | Evolutionary algorithm |
| $EV$ | Electric vehicle |
| $FC$ | Fuel cell |
| $GA$ | Genetic algorithm |
| $h$ | Height [m] |
| $HVAC$ | Heating, ventilation, and air conditioning |
| $I$ | Current [A] |
| $kNN$ | k-nearest neighbors |
| $LEM$ | Local energy management system |
| $\dot{m}$ | Mass flow [kg/s] |
| $m$ | Mass [kg] |
| $MAPE$ | Mean absolute percentage error |
| $MPC$ | Model predictive control |
| $MPP$ | Maximum power point tracking |
| $NRMSE$ | Normalized root mean square error |
| $P$ | Power [kW] |
| $p_c$ | Crossover probability [-] |
| $p_m$ | Mutation probability [-] |
| $PEM$ | Polymer electrolyte membrane |
| $PSO$ | Particle swarm optimization |
| $PV$ | Photovoltaic |
| $RC$ | Resistor–capacitor |
| $RES$ | Renewable energy sources |
| $RLF$ | Residential load factor |
| $SVM$ | Support vector machine |
| $SWT$ | Small Wind Turbine |
| $t_{life}$ | Lifecycle [a] |
| $v$ | Speed [m/s] |
| $\hat{y}$ | Predicted value [kW] |
| $y$ | Measured value [kW] |

## Appendix A. Calibration Results

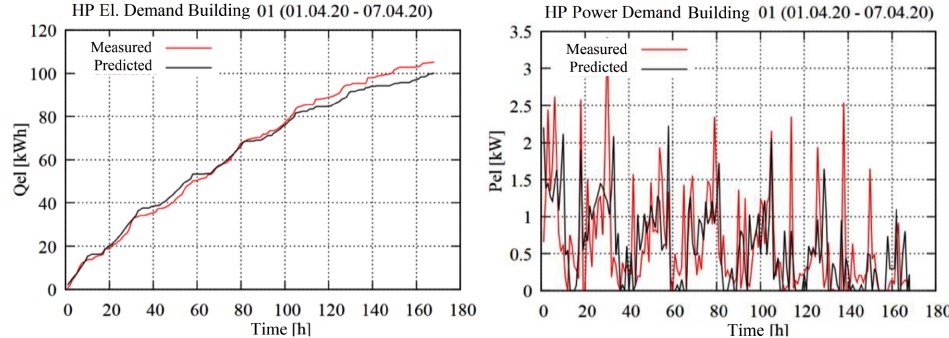

**Figure A1.** *Cont.*

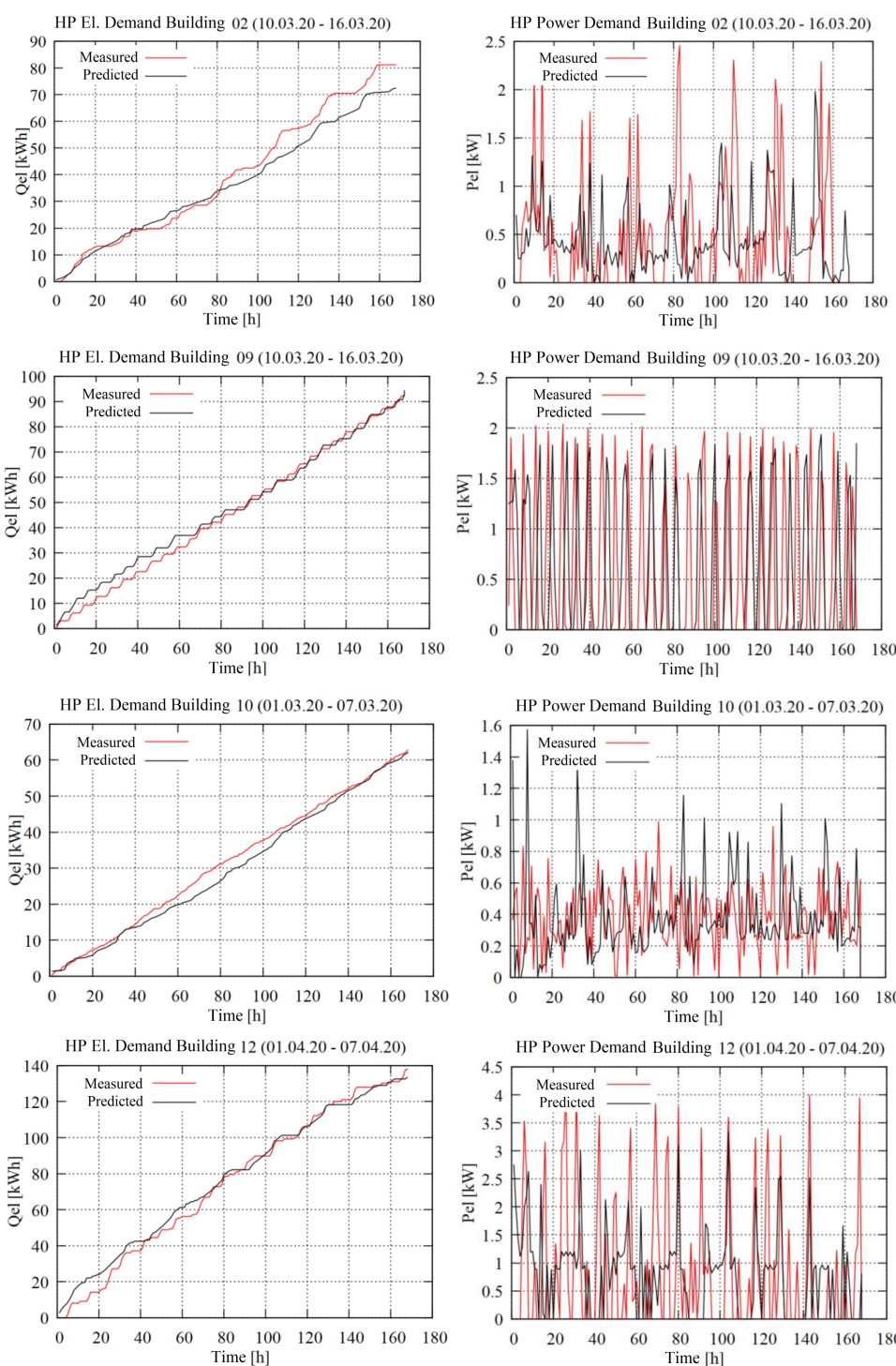

**Figure A1.** Building model calibration results (Building 1–12).

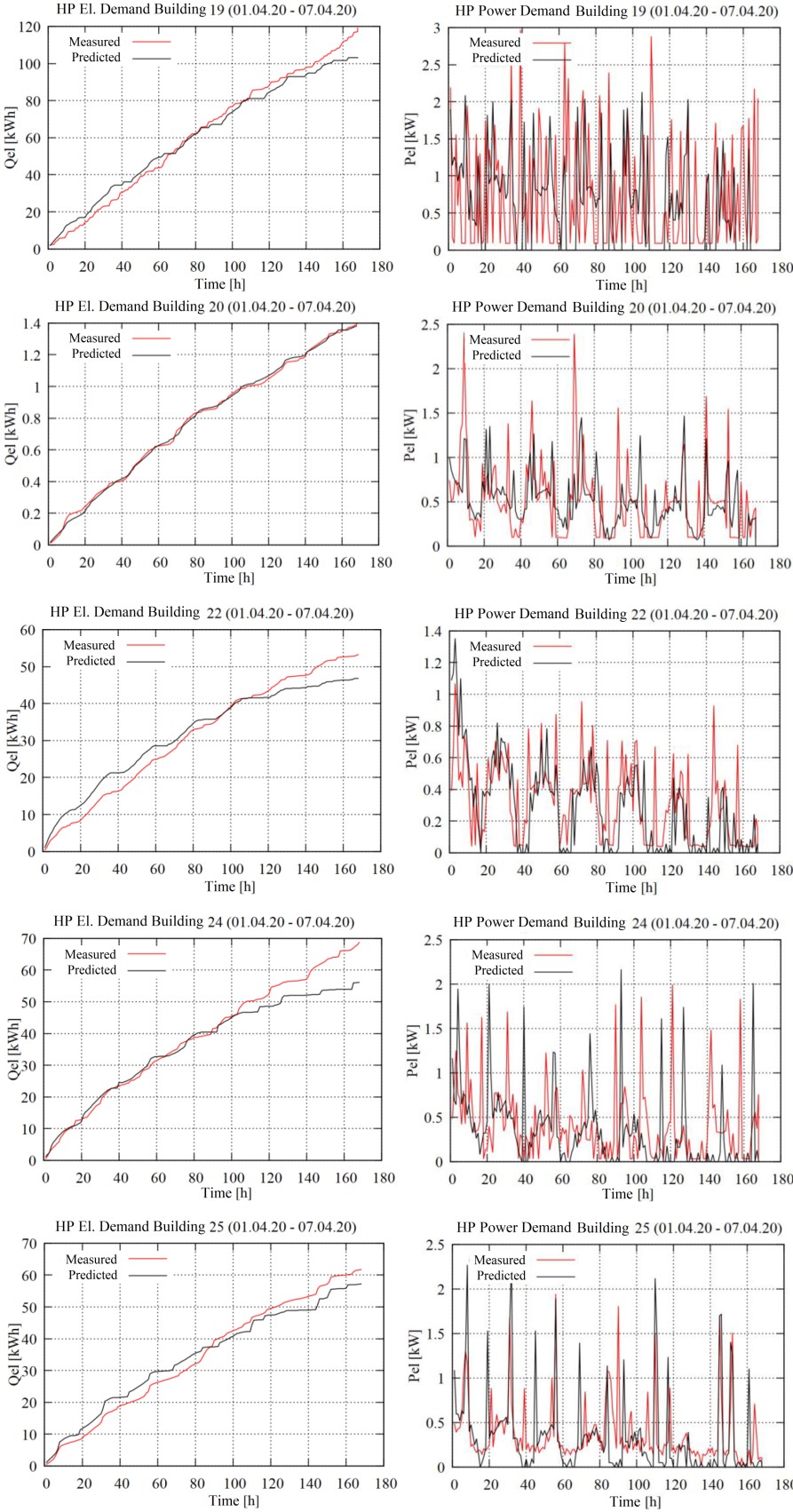

**Figure A2.** Building model calibration results (Building 19–25).

## Appendix B. Prediction Error Due to Weather Forecast Inaccuracy

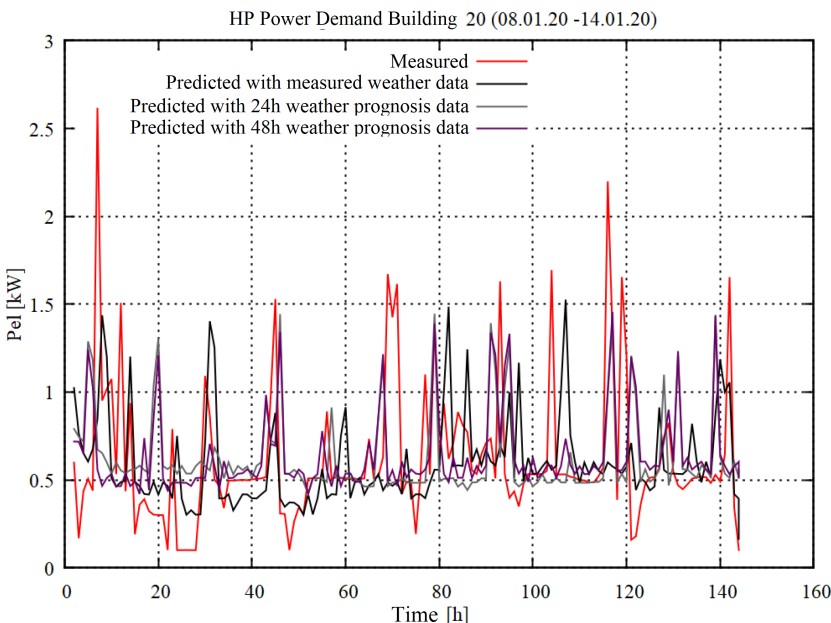

**Figure A3.** Impact of weather prediction inaccuracy on electrical heat pump power demand.

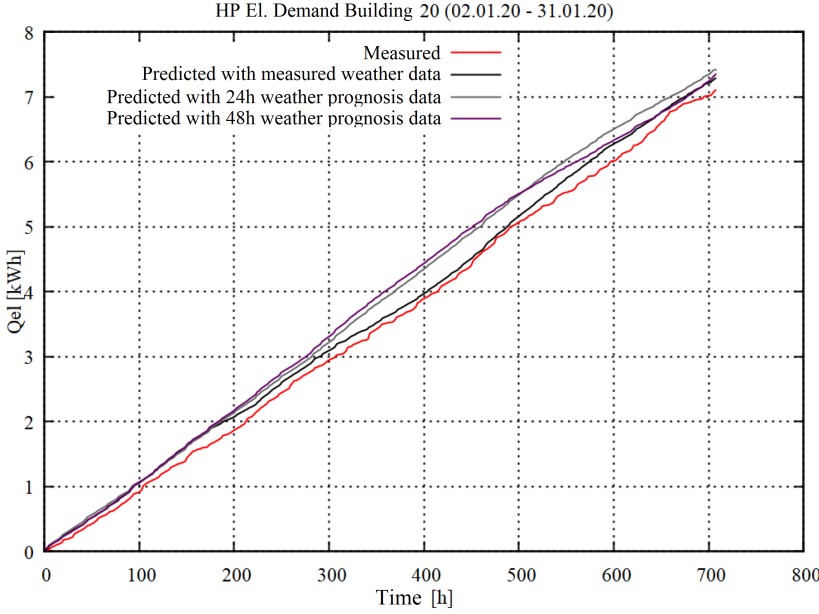

**Figure A4.** Impact of weather prediction inaccuracy on cumulative electrical heat pump energy demand.

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
