# Peer review of "Demand Side Management Based Power-to-Heat and Power-to-Gas Optimization Strategies for PV and Wind Self-Consumption in a Residential Building Cluster"

_energies, doi:10.3390/en14206712_

Round 1
Reviewer 1 Report
The topic of the article is relevant and interesting.
The article is generally well written but requires some minor changes. In the abstract, it is worth emphasizing the purpose of the article. The discussion would be much more interesting if the authors referred to other studies on this subject. Introduce a conclusion point, in which the Authors will present the degree of achievement of the goal, the main conclusion, research limitations and further research directions.
Author Response
- In the abstract, it is worth emphasizing the purpose of the article.
Answer: This has been adressed - The discussion would be much more interesting if the authors referred to other studies on this subject.
Answer: This has been added to the section “conclutions” - Introduce a conclusion point, in which the Authors will present the degree of achievement of the goal, the main conclusion, research limitations and further research directions
Answer: This section has been added
Reviewer 2 Report
The authors have carried out an interesting study and there is a significant quantity of work. However, the following points need to be taken care of before the work can be considered for publication, especially with regards to improving the clarity and quality of the presentation.
- Page 1, Line nos. 24 to 26: Please consider modifying this sentence - "To solve this problem sector coupling in the context of power-to-heat can be an important solution as it is predicted that heat pumps are gaining more and more importance within future heat supply systems" It is not clear in the grammatical usage. In fact, there are many sentences where clarity is lacking - it will be good if the authors can spend some more time and effort in improving the grammar, sentence formation and language.
- Page 8, Line nos. 271 to 273: "Application of a (genetic) optimization algorithm to the building parameters such as operating limit temperature, air exchange rate, room setpoint temperature, hysteresis of the buffer storage (heating and DHW) as well as the room heating hysteresis". Please provide detail range of parameters in the genetic optimization and results of optimization.
- Page 12, Line nos. 430 to 432: Please provide references for the selected number of computation- "Ten computational runs were performed for each parameter combination of the algorithm to reduce the effects of randomness, which occurs primarily when the number of generations and the population size are small". Is it enough to reduce the randomness? Is it too big for large number of generations and the populations?
- Fig. 3 and Fig. 4: Please consider adding a color scale to make it clearer. The best results region can be marked on the figure.
- Page 15, Line nos. 482 to 485: Please try to analysis why there is a great improvement and significant decline. " it can be seen that within this month under good conditions improvements of up to 23% can be achieved. These are evened out by those days with less or no optimization potential. There can also be seen two days with a significant decrease of up to -15%." The more explanations also need to be added in the parts of optimization results.
- The paper mentioned that “Fuel cell provides additional power for the building heat pumps, as well as waste heat.” Some latest literatures can be included to enrich the state-of-the-art review. See “Sustainable Residential Micro-Cogeneration System Based on a Fuel Cell Using Dynamic Programming-Based Economic Day-Ahead Scheduling. ACS Sustainable Chemistry & Engineering”
- Language of some places is obscure. Please polish.
- The authors should well define the concept of “self-consumption”. Does it mean the power generated by PV or wind is used to generate heat or hydrogen?
Author Response
- Page 1, Line nos. 24 to 26: Please consider modifying this sentence - "To solve this problem sector coupling in the context of power-to-heat can be an important solution as it is predicted that heat pumps are gaining more and more importance within future heat supply systems" It is not clear in the grammatical usage. In fact, there are many sentences where clarity is lacking - it will be good if the authors can spend some more time and effort in improving the grammar, sentence formation and language.
Answer: This and other sentences were edited to make them better understandable. - Page 8, Line nos. 271 to 273: "Application of a (genetic) optimization algorithm to the building parameters such as operating limit temperature, air exchange rate, room setpoint temperature, hysteresis of the buffer storage (heating and DHW) as well as the room heating hysteresis". Please provide detail range of parameters in the genetic optimization and results of optimization.
Answer: A detail range of parameters and all optimization results regarding the parameters were added as table 5 and table 7 as well as an additional interpretation within the text.
- Page 12, Line nos. 430 to 432: Please provide references for the selected number of computation- "Ten computational runs were performed for each parameter combination of the algorithm to reduce the effects of randomness, which occurs primarily when the number of generations and the population size are small". Is it enough to reduce the randomness? Is it too big for large number of generations and the populations?
Answer: The number of ten computational runs was considered as a compromise between accuracy and computational demand. E.g. for the case of a low number of generations of 50 and a population size of 5 (crossover probability of 0.9 and mutation probability of 0.3) the standard deviation would be 8.2% for 5 computational runs and 4.1% in the case of 10 computational runs. This has been added to the text.
- Fig 3 and Fig. 4: Please consider adding a color scale to make it clearer. The best results region can be marked on the figure.
Answer: A color scale has been added to the figures.
- Page 15, Line nos. 482 to 485: Please try to analysis why there is a great improvement and significant decline. " it can be seen that within this month under good conditions improvements of up to 23% can be achieved. These are evened out by those days with less or no optimization potential. There can also be seen two days with a significant decrease of up to -15%." The more explanations also need to be added in the parts of optimization results.
Answer: A more detailed explanation has been added to the text referring to the different prediction inaccuracies for PV generation as well as DHW and household electricity demand.
- The paper mentioned that “Fuel cell provides additional power for the building heat pumps, as well as waste heat.” Some latest literatures can be included to enrich the state-of-the-art review. See “Sustainable Residential Micro-Cogeneration System Based on a Fuel Cell Using Dynamic Programming-Based Economic Day-Ahead Scheduling. ACS Sustainable Chemistry & Engineering”
Answer: More references have been added to the subsection “Seasonal Hydrogen Storage”
- Language of some places is obscure. Please polish.
Answer: We have proofread and edited the article to address this issue.
- The authors should well define the concept of “self-consumption”. Does it mean the power generated by PV or wind is used to generate heat or hydrogen?
Answer: An explanation has been added to the introduction